# LEGO-FL: Learning Heterogeneous Federated Models as a LEGO Assembly Games

Zeqi Leng [1]   Chunxu Zhang [* 1]   Guodong Long [2]   Bo Yang [* 1]

## Abstract

Just as LEGO pieces can be assembled into an unlimited variety of structures, heterogeneous federated learning (HFL) can be viewed as the assembly of diverse model components. Inspired by this analogy, we reformulate HFL as a LEGO-like assembly game. The central challenge in HFL lies in learning across heterogeneous model architectures, which hinders direct parameter sharing. To address this challenge, we propose to decompose models into a set of modular components—analogous to LEGO pieces and collaboratively learn these components across clients under predefined composition rules. Based on this perspective, we develop a novel federated learning framework, termed LEGO-FL, which enables flexible model construction while preserving collaborative learning. Extensive experiments validate the effectiveness of LEGO-FL under different heterogeneous settings and system scales.

## 1. Introduction

Federated learning (FL) has emerged as a promising paradigm for collaborative model training across decentralized devices while preserving data privacy (McMahan et al., 2017; Kairouz et al., 2021). However, deploying FL models on edge devices introduces practical challenges. Owing to heterogeneous hardware capabilities and usage patterns, edge devices often differ significantly in their computational and memory resources available for training and deployment. As a result, FL systems must support models of varying sizes while maintaining architectural compatibility, giving rise to heterogeneous federated learning (HFL) (Ye et al., 2023; Pfeiffer et al., 2023; Liu et al., 2024; Li et al.). In HFL, the objective is to jointly learn heterogeneous models within a federated setting, such that models of different capacities can be deployed to edge devices according to their computational constraints.

Recent efforts have begun to address the challenges of HFL. To bypass architectural mismatches, intuitive approaches attempt to exchange knowledge-level semantics through federated distillation (Wang et al., 2023a; Zhang et al., 2023; Wang et al., 2024c; Wu et al., 2024a), shared classification heads (Yi et al., 2023), and prototype-based learning (Tan et al., 2022). Concurrently, modular strategies have emerged, carving out local architectures via neuron-level pruning within rigid, predefined supernets (Diao et al., 2020; Horvath et al., 2021; Wang et al., 2024a; Liang et al., 2025). Yet, whether relying on superficial knowledge exchange or being restricted by a predefined global architecture, a fundamental limitation persists: these methods underutilize structural diversity and inherently fail to tailor diverse personalized models for individual clients.

Motivated by this, this paper rethinks the HFL training paradigm. **We propose a LEGO-style orchestration that shifts from neuron-level pruning to block-level reassembly.** We first freely decompose heterogeneous models into neural blocks. Treating them as LEGO pieces, we assemble these blocks into diverse architectures under predefined rules. By recognizing and bridging latent correlations across these designs, this approach avoids rigid templates and enables flexible architectural choices for local clients. With structural flexibility established, collaboratively optimizing these diverse architectures emerges as the subsequent requirement, which poses a twofold challenge: first, the combinatorial nature of the reassembled topologies renders standard parameter aggregation infeasible; Second, directly customizing personalized local models without incorporating global feature representations leads to the loss of crucial generalized information.

To actualize this ambitious concept and address the ensuing challenges, we introduce LEGO-FL, a powerful and highly flexible framework uniquely designed for HFL. The lifecycle of LEGO-FL unfolds across three pivotal phases: Candidate Architecture Generation, Global Consensus Con-

---

[1]School of Jilin University, College of Computer Science and Technology, China [2]Australian Artificial Intelligence Institute, FEIT, University of Technology Sydney. Correspondence to: Bo Yang <ybo@jlu.edu.cn>, Chunxu Zhang <zhangchunxu@jlu.edu.cn>.

*Proceedings of the 43rd International Conference on Machine Learning*, Seoul, South Korea. PMLR 306, 2026. Copyright 2026 by the author(s).

struction, and Global-to-Local Personalization. Initially, we construct Equivalence Sets of neural blocks based on neural representation similarity. To navigate the vast compositional space efficiently, we design a Center-Anchored Tree Search (CTSearch), governed by non-strict ordered constraints and coupled with a dynamic block completion strategy to guarantee path integrity. This phase yields a diverse pool of candidate architectures. Subsequently, to circumvent the prohibitive costs of iterative training, we employ a training-free consensus construction (TfreeC), accurately pinpointing an optimal consensus architecture from the candidate pool. Finally, embracing the philosophy of bottom-up model stitching, we introduce a personalized grafting (PG) method that gracefully anchors generalized global knowledge onto personalized local knowledge, ultimately culminating in a streamlined distillation protocol to transfer the personalized models.

Our main contributions are summarized as follows:

- **Novel Training Paradigm for HFL.** We introduce LEGO-FL, which rethinks HFL through a flexible, LEGO-style assembly strategy. The framework allows heterogeneous networks to be freely decomposed and reassembled, fully exploiting structural diversity.

- **Low-Overhead Knowledge Aggregation.** We devise a training-free consensus selection mechanism paired with a tailored bottom-Top grafting strategy. Our method rapidly identifies the optimal global architecture and integrates it with local knowledge, accelerating the entire pipeline.

- **Superior Performance.** Comprehensive evaluations on three benchmark datasets reveal that LEGO-FL sets a new state-of-the-art, delivering superior performance over established baselines.

## 2. Related Work

### 2.1. Model Heterogeneity in Federated Learning

The fundamental challenge of HFL is orchestrating collaborative training across devices with dimensionally heterogeneous parameters. To bypass this structural mismatch, an intuitive paradigm abstracts parameters into transferable knowledge. Federated Distillation (FD) aligns soft predictions using public datasets (Jeong et al., 2018; Itahara et al., 2023; Wu et al., 2024a; Wang et al., 2024c). To circumvent the performance degradation and privacy risks associated with public data, data-free KD employs GANs to facilitate knowledge transfer (Zhang et al., 2023). Concurrently, feature prototypes and shared classification heads provide alternative pathways for representation-level knowledge sharing (Liang et al., 2020; Yi et al., 2023).

Distinct from knowledge-based methods, modular approaches confront architectural heterogeneity directly. While conventional NAS struggles with client-specific personalization (Tan et al., 2019; Wu et al., 2019), sub-model extraction techniques address this by customizing local models via parameter sparsification and neuron pruning from a predefined global network (Diao et al., 2020; Horvath et al., 2021; Wang et al., 2024a; Liang et al., 2025). However, this confines the solution space to a homo-architectural paradigm, where all models must originate from the same structural family.

In this work, we propose a more flexible training paradigm. Moving beyond shallow knowledge exchange, LEGO-FL embraces the rich diversity of structural topologies. Unconstrained by predefined architectural backbones, LEGO-FL empowers the free decomposing of heterogeneous models and the reassembly of neural blocks, thereby crafting bespoke architectures tailored to each client.

### 2.2. Model Reassembly Technique

Model Reassembly (MR) (Yang et al., 2022) seeks to orchestrate competitive architectures by freely combining disjoint equivalent sets of neural blocks. The first application of MR in FL established a standard pipeline (Wang et al., 2023b): network partitioning, neural block search, network matching, and client updating. Here, the efficiency of the search algorithm acts as a critical step, directly determining candidate quality and ultimate performance. (Wang et al., 2024b) attempted to ensure candidate diversity via a search strategy anchored by random blocks under strict sequential constraints. Yet, this mechanism is not entirely foolproof. On the one hand, the candidate outcomes are influenced by the total candidate pool size; on the other hand, strict sequential constraints restrict combinatorial diversity. Moreover, to satisfy the storage and compute capacities of edge devices, (Liu et al., 2025) constrained the network partitioning phase, enabling only the reassembly of specific blocks.

In this paper, LEGO-FL adopts a more practical paradigm. It removes the artificial predefined rules during the model partitioning phase and leverages a search algorithm based on non-strict ordered constraints, thereby generating a vastly richer candidate pool. To tackle model heterogeneity and the Non-IID data challenge, LEGO-FL also introduces two unprecedented phases consisting of Global Consensus Selection and Global-to-Local Personalization.

## 3. LEGO Assembly for Heterogeneous Federated Learning

This section details the design of the proposed LEGO-FL framework, as depicted in Figure 1. We reformulate the HFL aggregation challenge as a three-stage mapping process.

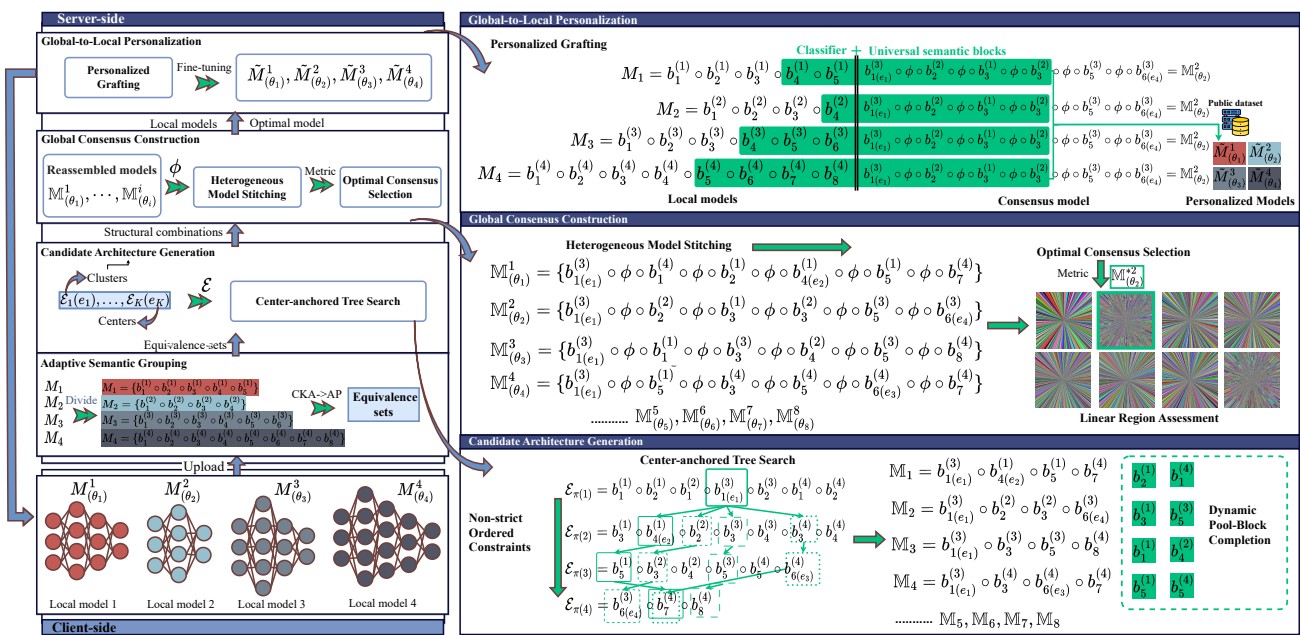

*Figure 1.* The overall workflow of LEGO-FL consists of three stages: (1) Candidate Architecture Generation: Generating diverse candidate architectures via Center-anchored Tree Search. (2) Global Consensus Construction: Selecting the optimal consensus model via the number of linear regions without iterative training. (3) Global-to-Local Personalization: Customizing local heterogeneous models through the personalized grafting approach.

Specifically, this process progressively constructs a global consensus from fragmented components and then derives customized, heterogeneous models for each client through a global-to-local personalization phase.

### 3.1. Problem Definition

**Federated Heterogeneous Setting.** Consider a FL system consisting of $N$ clients, denoted by a set $\mathcal{N} = \{1, 2, \ldots, N\}$. Each client $i \in \mathcal{N}$ holds a private local dataset $\mathcal{D}_i$ and maintains a heterogeneous local model $M_i$. The collection of these models forms a model zoo $\mathcal{Z} = \{M_i\}_{i=1}^N$. At each communication round $t$, a subset of clients $\mathcal{S}_t \subseteq \mathcal{N}$ is selected to participate in FL training.

**LEGO-Based Representation.** We conceptualize each heterogeneous local model $M_i$ as a sequence of functional units, which we term LEGO pieces (e.g., CNN layers or residual blocks). To enable collaboration across diverse architectures with varying depths and widths, we decompose these models into discrete LEGO pieces based on their feature abstraction levels. Let $b_k^i$ denote the $k$-th LEGO piece of client $i$. The model is formulated as:

$$M_i = b_{K_i}^i \circ b_{K_i-1}^i \circ \cdots \circ b_1^i, \quad (1)$$

where $K_i$ denotes the total number of pieces in client $i$.

To facilitate cross-architectural reassembly, we group these heterogeneous pieces into Equivalence Sets. We define $\mathcal{E}_l$ as the collection of pieces from participating clients that perform functionally equivalent feature extraction at the $l$-th semantic level:

$$\mathcal{E}_l = \left\{ b_k^i \mid \psi(b_k^i) = l, \ i \in \mathcal{S}_t, \ k \in \{1, \ldots, K_i\} \right\}, \quad (2)$$

where $\psi(\cdot)$ is a mapping function that categorizes heterogeneous pieces into $L$ global semantic levels. This flexible grouping allows LEGO pieces from different topological depths to be aligned within the same set $\mathcal{E}_l$ if they share similar representational roles.

**Definition 3.1** (Stage 1: Candidate Architecture Generation). The first objective is to construct a pool of topologically valid candidate models $\mathcal{A}_{\text{cand}}$ by assembling $L$ LEGO pieces under search spaces. Let $\pi(l)$ be a topological mapping function representing the valid semantic sequence, and $e_{\text{root}} \in \mathcal{E}_{\pi(1)}$ be the center anchor block. The generation process explores a constrained combinatorial subspace:

$$\mathcal{A}_{\text{cand}} = \left\{ z_{\pi(L)} \circ \cdots \circ z_{\pi(1)} \,\middle|\, z_{\pi(1)} = e_{\text{root}}, \ z_{\pi(l)} \in \mathcal{E}_{\pi(l)}, \forall l \right\}, \quad (3)$$

**Definition 3.2** (Stage 2: Consensus Construction). Given the generated candidate pool $\mathcal{A}_{\text{cand}}$, the second objective is to identify the single optimal global consensus model $G^*$. Instead of relying on computationally expensive iterative training, we seek a $G^*$ that maximizes a training-free performance proxy $P_{\mathcal{T}}(\cdot)$:

$$G^* = \arg\max_{A \in \mathcal{A}_{\text{cand}}} P_{\mathcal{T}}(A), \quad (4)$$

where $\mathcal{C}(\cdot)$ measures the architectural complexity. This step maps the heterogeneous inputs into a global consensus.

**Definition 3.3** (Stage 3: Global-to-Local Personalization). The final objective is to transfer the generalized knowledge from $G^*$ back to the heterogeneous clients. Following the philosophy of bottom-top model stitching, we decompose the global model into a feature extractor $F_g$ and a classifier $C_g$, such that $G^* = C_g \circ F_g$. Similarly, the local target model is decomposed as $M_i = C_i \circ F_i$.

We construct a personalized teacher model $\tilde{M}_i$ by grafting $F_g$ with the local classifier $C_i$. The local update on client $i$ minimizes the distillation loss with respect to its local parameters $\theta_i$:

$$\min_{\theta_i} \mathcal{L}_{\text{FD}}\left(M_i(x;\theta_i), \tilde{M}_i(x)\right), \quad \text{where } \tilde{M}_i = C_i \circ F_g. \tag{5}$$

Here, $\mathcal{L}_{\text{FD}}$ denotes the federated distillation loss that transfers knowledge from the local model $M_i$ with the personalized stitched teacher $\tilde{M}_i$.

### 3.2. Candidate Architecture Generation

This stage generates a pool of candidate architectures from heterogeneous local models through two steps: *Adaptive Semantic Grouping* to cluster blocks into Equivalence Sets, and *Center-Anchored Tree Search* (CTSearch) for efficient constraint-guided path exploration.

#### 3.2.1. ADAPTIVE SEMANTIC GROUPING

We first decompose local heterogeneous models into discrete blocks. While unrestricted decomposition ensures flexibility, establishing functional grouping rules is necessary. We cluster functionally similar blocks into disjoint Equivalence Sets based on representation similarity. Specifically, blocks producing similar output features for the same input are considered interchangeable. We use Centered Kernel Alignment (CKA) to quantify this similarity:

$$S(b, b') = \frac{1}{|\mathcal{D}_{\text{pub}}|} \sum_{x \in \mathcal{D}_{\text{pub}}} \text{CKA}\left(\mathbf{A}_b(x), \mathbf{A}_{b'}(x)\right), \tag{6}$$

where $\mathbf{A}_b(x)$ is the activation map of block $b$ for input $x$. A critical departure from conventional pipelines is our clustering strategy. Instead of the widely used K-means clustering, which is sensitive to the manually defined $K$ and struggles with varying model scales, we employ Affinity Propagation (AP) (Frey & Dueck, 2007). AP is non-parametric regarding the cluster count and directly yields actual block instances as cluster centers (exemplars), naturally benefiting the subsequent search phase.

#### 3.2.2. CENTER-ANCHORED TREE SEARCH

We formulate the search as a path graph problem $A = z_L \circ \cdots \circ z_1$, where $z_l \in \mathcal{E}_l$. To avoid the combinatorial explosion of random search, we propose CTSearch. For

**Center-Anchored Initialization,** the search is initialized using the shallowest exemplar $e_{\text{root}}$ identified by AP. Unlike prior methods that use random anchors, we fix the shallowest block because low-level semantic blocks exhibit minimal variance across heterogeneous architectures, providing a stable foundation.

**Search under Non-Strict Ordered Constraints.** To construct a valid and diverse network topology, our search algorithm operates on a dual-index system. First, at the macro level, we establish a global semantic order $\pi(l)$ by sorting all unique blocks in the model zoo. The search algorithm traverses the Equivalence Sets sequentially following this macro order (i.e., transitioning from $\mathcal{E}_{\pi(l)}$ to $\mathcal{E}_{\pi(l+1)}$). Second, at the micro level, we define $I(b)$ as the original depth index of block $b$ within its source heterogeneous model. While the algorithm moves to the immediate next semantic set $\mathcal{E}_{\pi(l+1)}$, it enforces a *non-strict ordered constraint* on the specific blocks to prevent invalid backward connections. Specifically, the next block $b_{\text{next}}$ is uniformly sampled from a filtered subset of $\mathcal{E}_{\pi(l+1)}$, where the original block indices must be strictly greater than the index of the current block $b_{\text{curr}}$. Formally:

$$b_{\text{next}} \sim \text{Uniform}\left(\left\{b \in \mathcal{E}_{\pi(l+1)} \mid I(b) > I(b_{\text{curr}})\right\}\right), \tag{7}$$

This dual-level formulation ensures that the reassembled architecture adheres to a logical semantic flow while accommodating the complex structural variations of local models.

**Dynamic Pool-Block Completion.** The depth tracking constraint $I(b) > I(b_{\text{curr}})$ can occasionally result in an empty or critically small valid subset (e.g., size $< \delta$), leading to premature termination. To prevent this, we introduce a dynamic fallback mechanism. If the valid subset at the next Equivalence Set is insufficient, the algorithm directly samples substitute blocks from all remaining unused blocks $(\mathcal{B}_{\text{total}} \setminus \mathcal{B}_{\text{curr}})$. This ensures the generation of complete network architectures while preserving candidate diversity.

### 3.3. Global Consensus Construction

Identifying the optimal consensus model $G^*$ from $\mathcal{A}_{\text{cand}}$ via standard iterative training is computationally prohibitive for FL systems. To address this, we introduce TfreeC, a zero-cost proxy method that evaluates architectural expressivity without weight optimization.

**Heterogeneous Model Stitching.** A candidate model $A = z_L \circ \cdots \circ z_1$ cannot be directly executed for evaluation if adjacent blocks from different clients have mismatched channel dimensions. To resolve this while adhering to the parameter constraints of edge devices, we, inspired by model stitching techniques, insert lightweight linear layers $\phi_l$ between blocks. These adapters align dimensions while approximating an identity mapping to preserve feature in-

formation. The stitched network is formulated as:

$$\hat{A} = z_L \circ \phi_{L-1} \circ \cdots \circ \phi_1 \circ z_1, \tag{8}$$

For a fair architectural evaluation, all weights in $\hat{A}$ are randomly initialized.

**Optimal Consensus Selection.** We evaluate the expressivity of $\hat{A}$ using Linear Region Complexity (Mellor et al., 2021). A network capable of dividing the input space into more linear regions possesses higher representational capacity. Given a small data batch $\mathbf{X} = \{x_1, \ldots, x_B\}$, we extract the binary activation matrix $\mathbf{C}_{\hat{A}} \in \{0, 1\}^{B \times H}$, where $H$ is the total number of ReLU activations. We compute the kernel matrix $\mathbf{K}_{\hat{A}} = \mathbf{C}_{\hat{A}} \mathbf{C}_{\hat{A}}^{\top}$ to capture activation similarities across the batch. The architecture's score is computed as:

$$S_{\text{score}}(\hat{A}) = \log \det \left( \mathbf{K}_{\hat{A}} + \epsilon \mathbf{I} \right), \tag{9}$$

where $\epsilon$ ensures numerical stability. The optimal consensus model is directly selected without training: $G^* = \arg\max_{\hat{A}} S_{\text{score}}(\hat{A})$.

### 3.4. Global-to-Local Personalization

The final stage transfers the consensus knowledge back to the heterogeneous clients. We propose *Personalized Grafting* (PG) via a "Bottom-Up Docking" mechanism, which constructs a hybrid teacher model by combining the global feature extractor with the local classifier.

**Grafting Construction.** We decompose the models into feature extractors and classifiers: the global consensus is $G^* = C_g \circ F_g$, and the local target model is $M_i = C_i \circ F_i$. To construct the personalized teacher $\tilde{M}_i$, we graft the global base $F_g$ with the local top $C_i$. A lightweight linear alignment layer $\phi_i$ is inserted to resolve dimensional mismatches:

$$\tilde{M}_i = C_i \circ \phi_i \circ F_g. \tag{10}$$

To minimize computational cost, the parameters of $F_g$ and $C_i$ are frozen. Only the alignment layer $\phi_i$ is optimized on a small public dataset $\mathcal{D}_{\text{pub}}$:

$$\min_{\phi_i} \mathbb{E}_{(x,y) \sim \mathcal{D}_{\text{pub}}} \left[ \mathcal{L}_{\text{CE}} \left( \tilde{M}_i(x; \phi_i), y \right) \right]. \tag{11}$$

**Knowledge Infusion via Distillation.** The grafted model $\tilde{M}_i$ serves as a teacher containing both global representations and local decision boundaries. To update the client model and achieve heterogeneous model delivery, we utilize the knowledge distillation loss proposed in (Wang et al., 2024b):

$$\begin{aligned}
\mathcal{L}_{\text{local}} = \frac{1}{|\mathcal{D}_i|} \sum_{(x,y) \in \mathcal{D}_i} &\Big[ \mathcal{L}_{\text{CE}}(M_i(x), y) \\
&+ \lambda \text{KL} \left( \sigma(M_i(x)) \parallel \sigma(\tilde{M}_i(x)) \right) \Big],
\end{aligned} \tag{12}$$

Here, $\lambda$ is a hyperparameter, KL denotes the Kullback-Leibler divergence, $\sigma(\cdot)$ is the softmax function. This formulation enables the local heterogeneous model to efficiently absorb global knowledge while maintaining personalization.

## 4. Experiments

### 4.1. Implementation Details

**Heterogeneous Model Zoos.** We design four heterogeneous model zoos with progressive complexity to evaluate the performance of our framework across diverse architectural landscapes. As shown in **Table 1**, the complexity progresses from 4 lightweight CNNs (Wang et al., 2023b) to a mix of CNNs and VGGs (Wang et al., 2024b). The system consists of $N = 50$ clients, each randomly assigned a model from the specified distributions.

**Datasets and Partitioning.** Experiments are conducted on **MNIST** (LeCun et al., 2010), **CIFAR-10** (Krizhevsky et al., 2009), and **SVHN** (Netzer et al., 2011) under both IID and Non-IID settings. For the Non-IID setup, we use a 2-class pathological label shift (Wang et al., 2023b), limiting each client to samples from two classes. Datasets are split 80%/20% for training/testing. A server-side public dataset containing 10% of the training data.

**Baselines.** We compare LEGO-FL with state-of-the-art FL methods. For Heterogeneous FL, the baselines include FedMD (Li & Wang, 2019), FedCache (Wu et al., 2024b), FedGH (Yi et al., 2023), pFedHR (Wang et al., 2023b), and pFedClub (Wang et al., 2024b). To demonstrate generalizability, we also compare with Homogeneous FL methods: FedAvg (McMahan et al., 2017), FedProx (Li et al., 2020), FLAYER (Chen et al., 2025), and DisUE (Leng et al., 2025).

**Configurations.** The framework is implemented in PyTorch and runs on an NVIDIA RTX 6000 GPU. We train for 100 communication rounds with $N = 50$ clients and a 0.1 participation ratio. Local updates use SGD for $E = 10$ epochs with a learning rate $\eta = 0.001$. The consensus candidate pool size is set to twice the number of active clients. All results are averaged over three independent runs with different random seeds.

### 4.2. Performance Comparison

**Heterogeneous Model Aggregation. (Main Task)** Table 2 compares classification accuracy across four model zoos. LEGO-FL consistently achieves state-of-the-art results under both IID and Non-IID settings. The results reveal three main findings: (1) Distillation methods (FedMD, FedCache) degrade as architectural heterogeneity increases (Zoo 1 to Zoo 4). Conversely, LEGO-FL remains robust. This indicates that our TfreeC approach constructs a more effective global representation than the representation-level align-

*Table 1.* Composition and characteristics of Heterogeneous Model Zoos. This table details the heterogeneity level and the number of clients assigned to each architecture.

| Zoo | Heterogeneity Level | CNN1 | CNN2 | CNN3 | CNN4 | VGG11 | VGG13 | VGG16 | Total |
|-----|---------------------|------|------|------|------|-------|-------|-------|-------|
| 1 | **Low** (4 lightweight CNNs) | 12 | 12 | 12 | 14 | - | - | - | 50 |
| 2 | **Medium** (Adds VGG11/16) | 10 | - | 10 | 10 | 10 | - | 10 | 50 |
| 3 | **Medium** (Adds VGG13) | 8 | - | 8 | 8 | 8 | 8 | 10 | 50 |
| 4 | **High** (Full Mix of 7 Models) | 7 | 7 | 7 | 7 | 7 | 7 | 8 | 50 |

*Table 2.* **Comparison with state-of-the-art methods.** Accuracy (%) on MNIST, CIFAR-10, and SVHN under IID and Non-IID settings across different model zoos. Red and blue indicate the best and second-best performance, respectively.

| Model Zoo | Method | MNIST | | CIFAR-10 | | SVHN | |
|-----------|--------|-------|---------|----------|---------|------|---------|
| | | IID | Non-IID | IID | Non-IID | IID | Non-IID |
| **Model Zoo 1** Low (4 CNNs) | FedCache | $93.52_{\pm 0.55}$ | $92.96_{\pm 1.13}$ | $68.06_{\pm 0.75}$ | $64.01_{\pm 1.68}$ | $81.83_{\pm 0.62}$ | $77.82_{\pm 0.85}$ |
| | FedMD | $92.16_{\pm 1.32}$ | $91.37_{\pm 1.56}$ | $67.14_{\pm 1.67}$ | $63.50_{\pm 1.88}$ | $80.22_{\pm 1.59}$ | $76.14_{\pm 1.86}$ |
| | FedGH | $92.93_{\pm 1.52}$ | $91.44_{\pm 1.08}$ | $67.88_{\pm 1.75}$ | $70.77_{\pm 1.93}$ | $79.03_{\pm 1.44}$ | $75.28_{\pm 1.75}$ |
| | pfedHR | $92.25_{\pm 1.93}$ | $91.07_{\pm 1.64}$ | $72.45_{\pm 1.81}$ | $69.08_{\pm 1.95}$ | $81.88_{\pm 2.36}$ | $79.25_{\pm 1.71}$ |
| | pfedClub | $93.24_{\pm 1.36}$ | $92.66_{\pm 0.98}$ | $74.88_{\pm 2.02}$ | $71.94_{\pm 1.82}$ | $82.69_{\pm 1.61}$ | $81.21_{\pm 1.68}$ |
| | **LEGO-FL (Ours)** | $96.66_{\pm 1.32}$ | $95.99_{\pm 1.08}$ | $78.33_{\pm 2.30}$ | $77.35_{\pm 2.10}$ | $86.02_{\pm 1.83}$ | $85.67_{\pm 0.71}$ |
| **Model Zoo 2** Medium (+VGG11/16) | FedCache | $92.82_{\pm 1.70}$ | $91.84_{\pm 0.87}$ | $68.56_{\pm 1.15}$ | $62.40_{\pm 0.74}$ | $82.16_{\pm 1.12}$ | $75.53_{\pm 1.14}$ |
| | FedMD | $91.09_{\pm 2.02}$ | $90.45_{\pm 2.69}$ | $66.08_{\pm 2.90}$ | $62.22_{\pm 2.33}$ | $79.06_{\pm 2.06}$ | $74.14_{\pm 1.69}$ |
| | FedGH | $92.55_{\pm 0.91}$ | $93.90_{\pm 1.22}$ | $73.83_{\pm 1.07}$ | $72.96_{\pm 1.14}$ | $82.83_{\pm 1.27}$ | $80.79_{\pm 1.59}$ |
| | pfedHR | $93.25_{\pm 1.93}$ | $92.07_{\pm 2.46}$ | $74.45_{\pm 2.80}$ | $70.08_{\pm 2.37}$ | $82.80_{\pm 2.45}$ | $81.25_{\pm 1.93}$ |
| | pfedClub | $94.56_{\pm 1.43}$ | $93.26_{\pm 1.99}$ | $76.35_{\pm 2.33}$ | $72.99_{\pm 2.33}$ | $84.28_{\pm 1.69}$ | $82.63_{\pm 1.54}$ |
| | **LEGO-FL (Ours)** | $97.25_{\pm 0.73}$ | $95.47_{\pm 1.55}$ | $80.82_{\pm 1.82}$ | $78.96_{\pm 1.31}$ | $87.70_{\pm 1.93}$ | $86.69_{\pm 1.20}$ |
| **Model Zoo 3** Medium (+VGG13) | FedCache | $92.16_{\pm 1.18}$ | $90.04_{\pm 1.96}$ | $67.04_{\pm 1.34}$ | $63.68_{\pm 1.85}$ | $80.58_{\pm 1.72}$ | $74.65_{\pm 1.56}$ |
| | FedMD | $90.25_{\pm 2.23}$ | $87.27_{\pm 2.37}$ | $65.46_{\pm 2.79}$ | $62.04_{\pm 2.58}$ | $78.52_{\pm 2.55}$ | $72.36_{\pm 2.33}$ |
| | FedGH | $93.77_{\pm 1.22}$ | $80.48_{\pm 0.16}$ | $75.61_{\pm 1.27}$ | $74.54_{\pm 0.95}$ | $81.23_{\pm 1.88}$ | $73.45_{\pm 1.53}$ |
| | pfedHR | $93.04_{\pm 2.74}$ | $90.39_{\pm 2.93}$ | $74.15_{\pm 2.65}$ | $69.33_{\pm 2.14}$ | $82.32_{\pm 2.64}$ | $80.38_{\pm 2.62}$ |
| | pfedClub | $94.62_{\pm 1.31}$ | $92.29_{\pm 1.29}$ | $75.67_{\pm 2.64}$ | $74.55_{\pm 2.41}$ | $84.08_{\pm 1.26}$ | $82.45_{\pm 1.19}$ |
| | **LEGO-FL (Ours)** | $97.05_{\pm 1.88}$ | $95.49_{\pm 1.62}$ | $80.88_{\pm 2.25}$ | $79.30_{\pm 2.42}$ | $87.74_{\pm 2.33}$ | $86.66_{\pm 2.24}$ |
| **Model Zoo 4** High (Mix of 7) | FedCache | $90.73_{\pm 1.52}$ | $89.42_{\pm 1.57}$ | $67.88_{\pm 0.90}$ | $62.75_{\pm 0.83}$ | $79.62_{\pm 1.04}$ | $73.39_{\pm 0.72}$ |
| | FedMD | $90.21_{\pm 3.17}$ | $85.33_{\pm 3.01}$ | $65.06_{\pm 3.87}$ | $61.90_{\pm 4.47}$ | $78.88_{\pm 3.96}$ | $71.61_{\pm 4.07}$ |
| | FedGH | $80.39_{\pm 1.49}$ | $88.94_{\pm 1.44}$ | $65.83_{\pm 1.28}$ | $63.62_{\pm 1.09}$ | $79.71_{\pm 1.69}$ | $74.05_{\pm 1.39}$ |
| | pfedHR | $92.27_{\pm 2.93}$ | $90.58_{\pm 2.80}$ | $73.40_{\pm 2.69}$ | $70.37_{\pm 3.11}$ | $82.56_{\pm 2.28}$ | $79.91_{\pm 2.63}$ |
| | pfedClub | $94.40_{\pm 1.95}$ | $91.88_{\pm 1.89}$ | $75.70_{\pm 2.25}$ | $71.46_{\pm 2.55}$ | $84.23_{\pm 1.90}$ | $80.55_{\pm 1.76}$ |
| | **LEGO-FL (Ours)** | $97.81_{\pm 1.77}$ | $95.99_{\pm 1.44}$ | $80.15_{\pm 1.64}$ | $79.88_{\pm 2.26}$ | $87.52_{\pm 2.31}$ | $86.48_{\pm 2.33}$ |

ment used in prior distillation methods. (2) Under Non-IID settings, LEGO-FL shows stronger resilience than existing reassembly methods, outperforming pFedClub by ∼5.9% on SVHN (Zoo 4). This improvement is driven by our Personalized Grafting module, which preserves local classifiers to handle label shifts while leveraging global features, avoiding the pitfalls of structural matching. (3) LEGO-FL significantly outperforms FedGH (e.g., +16.26% on CIFAR-10 Non-IID, Zoo 4). This demonstrates that sharing only classification heads is insufficient for deep heterogeneous networks. Our block-level reassembly allows comprehensive parameter sharing within the feature extractor, fully exploiting collaborative training.

**Homogeneous Model Aggregation.** We evaluate LEGO-FL in homogeneous settings to further investigate the efficacy of the reassembly mechanism within uniform architectures. It outperforms traditional FL baselines, indicating that the reassembly mechanism itself effectively generates diverse and competitive candidate structures. Moreover, it surpasses personalization-focused methods such as DisUE and FLAYER. This confirms that LEGO-FL achieves a better trade-off between global knowledge aggregation and local personalization.

**Scalability to Large-Scale Federation.** We investigate the scalability of our model under large-scale client scenarios. Specifically, we scale the federation to $N = 100$

clients while maintaining a fixed per-round participation budget of 10 active clients. This sparse participation setting is highly challenging for heterogeneous model training. Table 4 shows that LEGO-FL scales better than all baselines, outperforming the runner-up (pFedClub) by 5.36% on CIFAR-10. This advantage confirms that our global consensus design is significantly more effective for knowledge aggregation than standard similarity-matching approaches.

## 4.3. Ablation Study

We conduct a comprehensive ablation study to validate the necessity of the three key components in LEGO-FL. Specifically, we design a series of variants to evaluate every detail of our proposed pipeline. All ablation experiments are conducted on the CIFAR-10 dataset under the Non-IID setting. The results are summarized in Table 5.

**Effectiveness of CTSearch.** Macroscopically, the performance of LEGO-FL is heavily dictated by search efficiency, as the generated candidates fundamentally bound the final outcome. Microscopically, we highlight several notable findings: First, replacing the fixed root anchor with a randomly selected one degrades performance. This confirms that the selection of the first anchor is crucial, and fixing the shallowest semantic block provides a stable foundation for a topologically reasonable search path. Second, reverting to unconstrained random search not only compromises search efficiency but also degrades accuracy. Random search fails to guarantee path integrity, allowing poorly constructed architectures to participate in the training process. Finally, removing the *Dynamic Pool-Block Completion* strategy causes a significant performance drop, verifying the risk of premature termination during the search phase. Furthermore, restricting the completion scope strictly to ordered constraints also harms the results. This finding is intrinsically linked to our *non-strict ordered constraints*: to guarantee path integrity under relaxed sequential rules, expanding the completion scope to the global unused pool is necessary to compensate for search bottlenecks.

**Effectiveness of TFreeC.** To evaluate the consensus selection phase, we examine two critical variants. First, we replace the global consensus mechanism with a direct structural matching strategy (i.e., directly assigning the top-scoring candidates to clients based on local architecture counts). This performance drop indicates that a universally high-scoring architecture does not necessarily match local heterogeneous topologies directly. Furthermore, it corroborates that direct matching paradigms inevitably discard crucial global information. Second, we explore alternative zero-cost proxies. The results demonstrate, for the first time, that the *NASWOT* metric is highly effective and well-suited for evaluating representational capacity in heterogeneous federated learning systems.

**Effectiveness of PG.** We ablate the grafting mechanism by discarding the decoupling of the global and local models. Instead, we directly transfer the global consensus knowledge to the clients via standard knowledge distillation, a variant that essentially degenerates into a conventional federated distillation algorithm. The resulting performance degradation underscores the principle of HFL: the ultimate objective is to customize bespoke models tailored to specific clients, rather than merely exploiting generalized global information.

*Table 3.* **Performance on Homogeneous Settings.** Comparison of test accuracy (%) against homogeneous FL baselines on MNIST, CIFAR-10, and SVHN under Non-IID partitions.

| Model | Method | MNIST | CIFAR-10 | SVHN |
|---|---|---|---|---|
| CNN4 | FedAvg | $95.25_{\pm0.59}$ | $73.90_{\pm0.55}$ | $83.18_{\pm0.44}$ |
| | FedProx | $95.74_{\pm1.67}$ | $73.22_{\pm0.34}$ | $81.38_{\pm0.25}$ |
| | FLAYER | $95.90_{\pm0.22}$ | $75.28_{\pm0.40}$ | $86.21_{\pm0.16}$ |
| | DisUE | $97.07_{\pm0.05}$ | $76.26_{\pm0.46}$ | $86.20_{\pm0.49}$ |
| | **LEGO-FL** | $96.59_{\pm0.43}$ | $77.84_{\pm0.43}$ | $85.92_{\pm0.52}$ |
| VGG11 | FedAvg | $76.67_{\pm0.34}$ | $66.87_{\pm0.85}$ | $70.86_{\pm0.30}$ |
| | FedProx | $72.94_{\pm0.24}$ | $66.89_{\pm0.24}$ | $70.26_{\pm0.45}$ |
| | FLAYER | $76.58_{\pm0.42}$ | $70.08_{\pm0.09}$ | $73.83_{\pm0.83}$ |
| | DisUE | $77.42_{\pm0.46}$ | $70.12_{\pm0.77}$ | $74.64_{\pm0.47}$ |
| | **LEGO-FL** | $79.33_{\pm0.48}$ | $72.66_{\pm0.32}$ | $75.86_{\pm0.56}$ |

## 4.4. Computational Overhead Analysis

Computational overhead remains a critical technical bottleneck for model reassembly methods. To demonstrate the efficiency of LEGO-FL, we systematically evaluate the computational costs of its three core stages: clustering, consensus selection, and personalization. We then benchmark these costs against the state-of-the-art reassembly method, pFedClub (Wang et al., 2024b). Macro-level results indicate that LEGO-FL drastically reduces the overall computational overhead, a massive advantage inherently derived from our training-free consensus mechanism and lightweight knowledge grafting approach.

Specifically, **Figure 2a** illustrates the cost comparison dur-

*Table 4.* **Scalability Analysis.** Comparison with baselines on MNIST, CIFAR-10, and SVHN under a large-scale setting ($N = 100$ clients, Non-IID).

| Zoo | Method | MNIST | CIFAR-10 | SVHN |
|---|---|---|---|---|
| Model Zoo 1 ($N = 100$) | FedCache | $85.45_{\pm1.38}$ | $63.71_{\pm1.06}$ | $75.21_{\pm1.36}$ |
| | FedMD | $86.57_{\pm1.42}$ | $60.73_{\pm1.62}$ | $73.78_{\pm1.11}$ |
| | FedGH | $77.09_{\pm1.98}$ | $66.14_{\pm0.82}$ | $76.51_{\pm1.93}$ |
| | pFedHR | $86.54_{\pm1.85}$ | $66.98_{\pm1.96}$ | $73.12_{\pm2.20}$ |
| | pFedClub | $87.15_{\pm2.23}$ | $67.74_{\pm1.59}$ | $77.52_{\pm2.38}$ |
| | **LEGO-FL** | $89.86_{\pm0.89}$ | $73.10_{\pm1.72}$ | $79.07_{\pm1.98}$ |

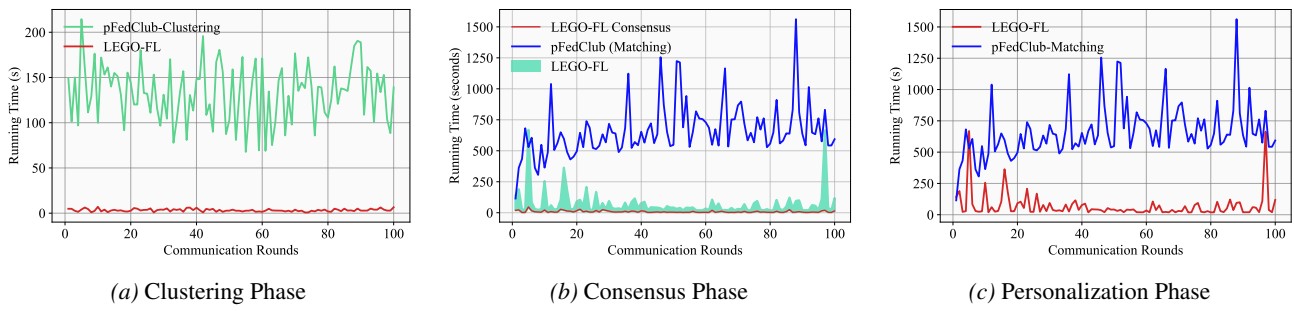

*(a)* Clustering Phase      *(b)* Consensus Phase      *(c)* Personalization Phase

*Figure 2.* **Computational Overhead Analysis.** Comparison of time costs across three stages between LEGO-FL and pFedClub. **(a)** The AP clustering algorithm converges faster than traditional methods. **(b)** The training-free consensus selection incurs negligible latency. **(c)** Our model-stitching-based knowledge grafting is significantly more efficient than structural matching.

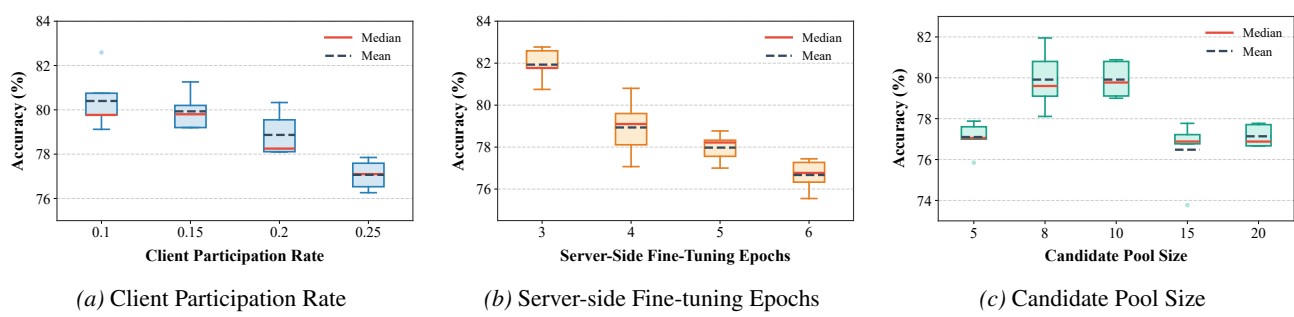

*(a)* Client Participation Rate      *(b)* Server-side Fine-tuning Epochs      *(c)* Candidate Pool Size

*Figure 3.* **Hyperparameter Sensitivity Analysis.** Impact of three critical hyperparameters on the classification accuracy of LEGO-FL. **(a)** Increasing the client participation rate degrades performance. **(b)** Excessive server-side fine-tuning introduces unnecessary bias. **(c)** A candidate pool size of approximately 10 provides optimal structural diversity.

*Table 5.* **Ablation Study.** Impact of key components (CTSearch, TFreeC, PG) on CIFAR-10 classification accuracy (%) under IID and Non-IID settings.

| Module | Variant / Modification | IID | Non-IID |
|---|---|---|---|
| **CTSearch** | Anchor: Fixed $\rightarrow$ Random | 76.01 | 75.55 |
| | Search: Ordered $\rightarrow$ Random | 74.21 | 73.33 |
| | Completion: w/o Completion | 75.85 | 74.20 |
| | Completion: Random $\rightarrow$ Ordered | 73.46 | 71.33 |
| **TFreeC** | Consensus: Unique $G^*$ $\rightarrow$ Top-$N$ | 74.48 | 72.26 |
| | Metric: NASWOT $\rightarrow$ Jacov | 79.28 | 79.00 |
| | Metric: NASWOT $\rightarrow$ Fisher | 78.22 | 77.77 |
| **PG** | w/o PG (Global Distillation only) | 78.55 | 77.16 |
| **Full** | **LEGO-FL (Full Pipeline)** | **80.60** | **79.12** |

ing the clustering phase. The results confirm that our adopted Affinity Propagation (AP) algorithm is suited for HFL systems, achieving significantly faster convergence than traditional methods. **Figure 2b** demonstrates that although LEGO-FL introduces an additional Global Consensus Selection stage, this process is highly efficient due to the zero-cost proxy, adding virtually negligible latency to the pipeline. Finally, **Figure 2c** compares our personalization stage against the proprietary network matching phase

of pFedClub. During the dissemination of personalized models, our model-stitching-based knowledge grafting exhibits vastly superior efficiency compared to the structural matching relied upon by prior works.

### 4.5. Hyperparameter Sensitivity Analysis

We evaluate the robustness of LEGO-FL with respect to three critical hyperparameters: the client participation rate, the number of server-side fine-tuning epochs, and the total number of reassembled candidate models. The empirical results are visualized in **Figure 3**.

**Client Participation Rate.** As illustrated in **Figure 3(a)**, unlike traditional studies focusing solely on statistical heterogeneity, the performance of our framework remains largely stable, validating the reliability of the architecture search phase. However, a slight performance degradation occurs as the participation rate increases. This is intuitive: a larger pool of active clients introduces greater architectural variance, which escalates the difficulty of collaborative training.

**Server-Side Fine-Tuning Epochs.** As demonstrated in **Figure 3(b)**, regarding the optimization of the alignment layer for the personalized teacher, we observe that increasing the number of fine-tuning epochs paradoxically degrades the performance of LEGO-FL. This suggests that over-tuning

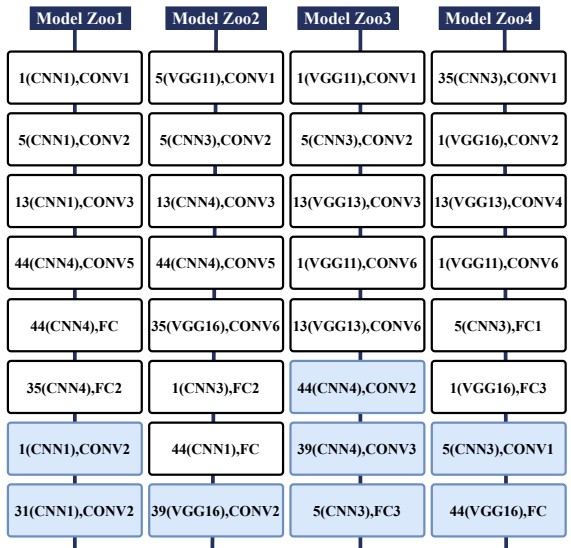

*Figure 4.* **Visualization of Reassembled Architectures.** Examples of candidate topologies generated by our Center-Anchored Tree Search across four model zoos. Guided by non-strict ordered constraints (white blocks) and dynamic block completion (blue blocks), LEGO-FL discovers diverse, cross-architectural paths.

the lightweight connector on a public dataset leads to over-fitting, which introduces unnecessary bias and disrupts the generalized features.

**Candidate Pool Size.** As depicted in **Figure 3(c)**, regarding the total number of reassembled candidate models, we discover an interesting phenomenon: generating a candidate pool size approximately twice the number of active clients is sufficient to capture adequate structural diversity. This optimal threshold ensures that a highly competitive consensus architecture can be identified without demanding an excessively large search space.

### 4.6. Visualization of Reassembled Architectures

To provide intuitive insights into our Center-Anchored Tree Search algorithm tailored for HFL, we visualize a subset of the reassembled candidate architectures in **Figure 4**. Notably, guided by the non-strict ordered constraints and the dynamic pool-block completion mechanism, LEGO-FL simultaneously guarantees the integrity of the search paths to yield topologically valid architectures, while ensuring exceptional diversity in the generated outcomes. This flexibility allows the framework to transcend architectural boundaries, fully exploiting the structural richness in local heterogeneous models.

### 5. Conclusion

This work proposes LEGO-FL, a novel training paradigm tailored explicitly for HFL. To better customize personalized

heterogeneous models for local clients, LEGO-FL shifts the fundamental methodology from restrictive neuron-level model pruning to highly flexible block-level reassembly. Furthermore, during the knowledge aggregation process, our framework introduces a global-to-local personalization mechanism that effectively balances global commonality with local specificities. Consequently, LEGO-FL not only resolves the challenge of heterogeneous model aggregation but also simultaneously mitigates the pervasive issue of data heterogeneity (Non-IID data).

### Acknowledgements

Zeqi Leng, Chunxu Zhang and Bo Yang are supported by the National Natural Science Foundation of China under Grant Nos. U22A2098, 62206105, and 62202200; the Major Science and Technology Development Plan of Jilin Province under Grant No.20240212003GX, the Major Science and Technology Development Plan of Changchun under Grant No.2024WX05.

### Impact Statement

The proposal of LEGO-FL represents a significant milestone in Heterogeneous Federated Learning (HFL), effectively resolving the challenge of heterogeneous model aggregation under Non-IID data distributions. Analogous to LEGO bricks, LEGO-FL champions the free decomposition of local heterogeneous models, followed by dynamic reassembly guided by an efficient search strategy tailored explicitly for HFL. Notably, aligning with the latest trends, LEGO-FL orchestrates a training-free global consensus evaluation and a consensus-to-personalization phase, striking an optimal balance between generalized and personalized knowledge. Furthermore, it drastically slashes the computational overhead that bottlenecks reassembly-based algorithms, thereby paving the way for the broader adoption of model reassembly paradigms in FL.

Despite offering an effective modular paradigm, LEGO-FL presents two limitations. First, the selection of a single global consensus may inadvertently discard certain localized structural information. In future work, we plan to investigate enhanced consensus selection strategies or construct augmented consensus models to ensure that the global representation captures a more comprehensive spectrum of local information. Second, while LEGO-FL takes a monumental step towards the ultimate goal of model reassembly, it still inevitably relies on distillation protocols as an intermediary to transfer knowledge to heterogeneous clients. Moving forward, we aim to design a radically novel knowledge transfer paradigm for HFL, with the ambition of completely breaking free from the constraints of knowledge distillation and approaching the ultimate ideal of pure model reassem-

bly. Finally, from our perspective, the proposed framework introduces no negative social impacts.

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
