# OpenReview forum: "LEGO-FL: Learning Heterogeneous Federated Models as a LEGO Assembly Games"
_ICML.cc/2026/Conference — ICML 2026 regular_

### Official Review · Reviewer_BWiG · 2026-03-03

**Soundness:** 2
**Presentation:** 2
**Significance:** 2
**Originality:** 3
**Overall Recommendation:** 3
**Confidence:** 4

**Summary:**

In this paper, the authors propose LEGO-FL, a framework for heterogeneous federated learning. The approach formulates the problem of model heterogeneity as an assembly task, treating neural network blocks similarly to Lego pieces. Specifically, LEGO-FL introduces a three-stage process involving Adaptive Semantic Grouping, Center Anchored Tree Search, Training Free Consensus Selection using NASWOT scores, and Personalized Grafting. The method aims to build a global consensus model and effectively adapt it to local clients. Experiments conducted on standard datasets demonstrate improvements over existing heterogeneous federated learning baselines.

**Compliance With Llm Reviewing Policy:**

Affirmed.

**Final Justification:**

While I thank the authors for their detailed clarifications, my fundamental concerns remain inadequately resolved. Specifically, the bounding layer depth fails to account for the substantial FLOP and parameter discrepancies across heterogeneous blocks. Furthermore, the method's robustness to distribution-shifted public datasets lacks empirical validation. Therefore, I will maintain my original score.

**Key Questions For Authors:**

Please see the Weaknesses above.

**Limitations:**

1. The empirical evaluation is restricted to smaller-scale datasets like CIFAR-10 and traditional architectures like CNN, leaving the scalability to modern large-scale models unverified.
2. The reliance on random linear projections $\phi_l$ to bridge dimensional mismatches lacks robust theoretical justification regarding the preservation of representational capacity.
3. Valuating functional similarity purely via activation outputs ignores the potential computational complexity variations among heterogeneous blocks.

**Strengths And Weaknesses:**

Strengths:
1. The paper introduces an innovative three-stage framework for heterogeneous federated learning that effectively balances global consensus and local personalization without relying on a predefined super-model structure.
2. The proposed Training Free Consensus Selection significantly accelerates the process of finding the optimal global architecture $G^*$ by utilizing the linear region assessment metric $S_{score}(A)$ in Eq.(9) instead of costly iterative training.
3. The empirical results demonstrate strong performance and scalability across various model zoos and data distributions, outperforming recent baselines like pFedClub and FedGH while maintaining lower computational overhead.

Weaknesses:
1. The reliance on random linear projections $\phi_l$ in Eq.(8) to match channel dimensions during the Training Free Consensus Selection stage lacks rigorous theoretical backing. It is unclear how well this preserves the actual representational capacity of the assembled blocks.
2. The ablation study shows a performance drop when Dynamic Pool Block Completion is removed. This raises questions about the stability of the initial Adaptive Semantic Grouping phase and whether the clustering often fails to produce complete valid paths naturally.
3. The experimental setup is limited to relatively small datasets like CIFAR-10 and SVHN. It also relies primarily on standard CNN and VGG architectures, leaving the framework scalability to larger datasets and modern architectures unverified.
4. The paper claims a training-free approach for the consensus stage, but the Personalized Grafting phase still requires fine-tuning the connector $\phi_i$ on a server-side public dataset $\mathcal{D}\_{pub}$ via Eq.(12). Consequently, the performance of the method depends heavily on the selection of $\mathcal{D}\_{pub}$.
5. The metric for functional similarity, Centered Kernel Alignment, only considers activation outputs. This might completely ignore structural efficiency differences between blocks that produce similar outputs but have vastly different computational costs.
6. The font size for certain text in Figure 1 and Table 1 is excessively small, compromising readability.

---

> ### Author Rebuttal · Authors · 2026-03-31
>
> **Q1: [Weaknesses 1&Limitations 2] Function of the Linear Adapter Layer.**
> Thank you for your incisive observation. The linear adapter's role is simply to resolve dimensional mismatches. By learning an approximate identity mapping, it can achieve this while maximally preserving the feature information from the block's output. The effectiveness of this design has been validated in prior works [1, 2].
>
>
> **Q2: [Weaknesses 2] About Dynamic Pool Block Completion.**
> Clustering heterogeneous neural blocks into equivalence sets can be viewed as an imbalanced graph partitioning problem , and finding the optimal partition is NP-hard [3]. To prevent premature search termination, we designed the Dynamic Pool Block Completion mechanism to ensure the completeness of the generated model architectures.
>
> **Q3: [Weaknesses 3&Limitations 1] Experiments on Complex Datasets and Models.**
> Thank you for your constructive suggestion！ The datasets we initially used, MNIST, CIFAR-10, and SVHN, along with the CNNs and VGGs, are standard benchmarks widely adopted in prior works [1,2]. To further validate our method on more challenging tasks, we have supplemented our experiments on FEMNIST and CIFAR-100. The FEMNIST dataset is approximately 10x larger than CIFAR-10, and CIFAR-100 is a more difficult 100-class classification task. Furthermore, we selected more complex ResNet models, which are approximately 2x deeper than the CNNs and 1x deeper than the VGGs. As shown in Table 1, we assigned heterogeneous clients to ResNet-18 (26 clients), ResNet-34 (12 clients), and ResNet-50 (12 clients). Across all these settings, our method consistently outperforms the baseline methods. Furthermore, we have supplemented the results for our Model zoos (1-4) with their performance on the realistic datasets FEMNIST and Celeba (Celeba contains 200,288 samples, over 3x larger than CIFAR-10). Table 2 demonstrates the effectiveness of LEGO-FL.
>
> Table1:
> | Models | Methods | Cifar100 (noniid) | FEMNIST (noniid) |
> |:---|:---|:---:|:---:|
> | RESNETs| Fedcache | 31.48 | 67.60 |
> | RESNETs | FEDMD | 30.07 | 69.70 |
> | RESNETs | FedGH | 36.82 | 74.55 |
> | RESNETs | pFedHR | 35.37 | 76.21 |
> | RESNETs| pFedClub | 37.11 | 77.67 |
> | RESNETs | **Ours** | **39.97** | **78.00** |
>
> Table 2:
> | Zoos | Methods | FEMNIST (noniid) | Celeba (noniid) |
> |:---|:---|:---:|:---:|
> | Zoo1 | Ours | 71.25 | 85.15 |
> | Zoo2 | Ours | 73.33 | 87.92 |
> | Zoo3 | Ours | 73.73 | 86.85 |
> | Zoo4 | Ours | 75.11 | 86.22 |
>
> **Q4: [Weaknesses 4] Public Dataset Selection.**
> Thank you for pointing this out. The public dataset is used to fine-tune the personalized models, with the purpose of smoothly "connecting" the global feature extractor to the client's local classifier. Our method allows for considerable flexibility in the choice of the public dataset. Consistent with the approach in [1], we randomly sample 10% of the training data to serve as the public dataset.
>
> **Q5: [Weaknesses 5&Limitations 3] Functional Similarity.**
> Thank you for your insightful observation! Functional similarity is defined by the concept of representational similarity. The Centered Kernel Alignment (CKA) score between neural blocks is the indicator of this functional similarity. This is because CKA quantifies whether two neural blocks produce geometrically similar output activation spaces when given the same set of inputs. Specifically, CKA first computes the Gram matrix for each block's activation matrix. Each resulting kernel matrix captures the relational geometry of the input data within that block's respective activation space. A high CKA score—achieved by measuring the alignment of these centered and normalized kernel matrices—therefore indicates that the two blocks organize the data in an equivalent way.
>
> **Q6: [Weaknesses 6] Font Size Adjustment.**
> We will increase the font size in Figure 1 and Table 1 to ensure they are legible.
>
> **References**:
>
>  [1] Towards personalized federated learning via heterogeneous model reassembly[J]. NeurIPS.2023.
>
>  [2] pFedClub: Controllable heterogeneous model aggregation for personalized federated learning[J]. NeurIPS.2024.
>
>  [3] Deep model reassembly[J]. Advances in neural information processing systems, NeurIPS.2022.

---

> > ### Author Rebuttal · Reviewer_BWiG · 2026-04-02
> >
> > Thank you to the authors for the detailed rebuttal. However, several core methodological and theoretical concerns remain unresolved:
> >
> > **1. Contradiction regarding the Linear Adapter $\phi_l$**.
> > The rebuttal claims the adapter "learns an approximate identity mapping." This directly contradicts the manuscript (Section 3.3, Lines 242-244), which explicitly states $\phi_l$ is kept strictly *random* and untrained. The theoretical justification for how an untrained random projection preserves representational capacity during consensus selection remains unaddressed.
> >
> > **2. Computational Complexity**.
> > The response simply re-explains CKA's mechanism but evades the core issue: evaluating block similarity purely via activation outputs ignores structural efficiency. Treating a computationally heavy block and a lightweight block as interchangeable (if they yield high CKA scores) ignores the vastly different computational burdens placed on edge clients.
> >
> > **3. Privacy Constraints**.
> > The authors state that 10% of the training data is sampled for $\mathcal{D}_{pub}$. Centralizing training data fundamentally violates the strict privacy constraints of federated learning. Additionally, the method's robustness when using a truly independent, distribution-shifted public dataset was not evaluated.
> >
> > Given that these issues have not been adequately addressed, I will maintain my current score.

---

> > > ### Author Response · Authors · 2026-04-04
> > >
> > > **Q1: [Questions 1] Regarding the Linear Adapter and Representational Capacity.**
> > >
> > > Our method employs two distinct types of linear adapters, which serve different functions in Stage 2 (Training-Free Consensus Selection) and Stage 3 (Personalized Grafting).
> > >
> > > The “fine-tuned adapter” that learns an approximate identity mapping refers only to the adapter used in Stage 3. We apologize for any confusion in our previous response. In contrast, the adapter highlighted in the manuscript (Lines 242-244) belongs to Stage 2.
> > >
> > > Regarding the preservation of representational capacity, our explanation relies on a fundamental principle: the network’s capacity is dictated by non-linear activation functions, whereas our inserted adapter is a purely linear transformation. Specifically, we quantify a network’s representational capacity by the “number of linear regions.” In neural networks, these linear regions are formed by partitioning the feature space using non-linear activation functions, such as ReLU. Because the untrained adapter is a purely affine/linear transformation (y = Wx + b), it can only scale or rotate the feature space. It lacks the mathematical capacity to bend the space or create new decision boundaries. Therefore, inserting a linear adapter between blocks does not affect the total count of linear regions.
> > >
> > > **Q2: [Questions 2] Concern on Computational Complexity and Structural Efficiency.**
> > >
> > > We sincerely appreciate the reviewer for highlighting the critical issue of structural efficiency and the potential computational overhead imposed on clients. We agree that evaluating block similarity purely via CKA does not account for computational complexity. To explicitly address this and prevent the substitution of lightweight blocks with computationally heavy ones, our framework does not rely solely on CKA; rather, it incorporates an ordered-constrained search mechanism to govern the final personalized model’s size.
> > >
> > > Specifically, during the search phase, we retrieve the required blocks based on their block indices. Crucially, a layer depth constraint is introduced in this stage to prevent unbounded searching and excessive model growth. For instance, if a client’s local CNN comprises 4 to 6 layers, the search space for the global model is strictly bounded to 6 to 8 layers. Consequently, the final size and computational complexity of the generated personalized model remain closely aligned with the client’s original local architecture. Therefore, this mechanism mitigates computational burdens placed on edge clients.
> > >
> > > **Q3: [Questions 3]  Privacy Concern For Public Datasets .**
> > > We appreciate the reviewer’s rigorous evaluation regarding privacy and the distribution shift of the public dataset. First, regarding privacy constraints, sampling 10% of the training data for D_pub is an experimental simulation representing an accessible, open-source proxy dataset [1][2][3][4][5]. Such data partitioning is a common practice in federated learning. In our setup, this public dataset is completely disjoint from the clients' 90% private data. Because there is no exposure of local data, this setup strictly adheres to federated learning privacy guarantees.
> > >
> > > Second, regarding robustness against a distribution-shifted public dataset, the goal of fine-tuning in Stage 3 is not to learn new task-specific semantic features, but to perform architectural calibration. This step is crucial because the personalized model is assembled by grafting blocks from different local models, which inevitably leaves the feature spaces at the block boundaries misaligned. To address this gap, forward passes on D_pub act as excitation signals to harmonize these boundaries and smooth feature transitions. Consequently, as long as the public dataset shares a broad domain with the task (e.g., general natural images for visual tasks), it can effectively calibrate the grafted architecture.
> > >
> > > [1]Exploring the Distributed Knowledge Congruence in Proxy-data-free Federated Distillation. ACM Trans. Intell. Syst. Technol. 15, 2, Article 28 (April 2024), 34 pages. https://doi.org/10.1145/3639369
> > >
> > > [2]Heterogeneous Federated Learning via Model Distillation. NeurIPS.2019.
> > >
> > > [3]Distillation-based semi-supervised federated learning for communication-efficient collaborative training with non-iid private data,” IEEE Trans. Mobile Comput. vol. 22, no. 1, pp. 191–205, 2021.
> > >
> > > [4]FedBiKD: Federated Bidirectional Knowledge Distillation for Distracted Driving Detection,IEEE Internet Things J., vol. 10, no. 13, pp. 11643–11654, 2023.
> > >
> > > [5]Incentive-Based Federated Learning for Digital-Twin-Driven Industrial Mobile Crowdsensing. IEEE Internet Things J., vol. 10, no. 20, pp. 17851–17864, 2023.

---

### Official Review · Reviewer_hV7i · 2026-03-12

**Soundness:** 2
**Presentation:** 3
**Significance:** 3
**Originality:** 3
**Overall Recommendation:** 4
**Confidence:** 3

**Summary:**

This paper introduces LEGO-FL, which is a method for heterogeneous federated learning that is different from prior methods in the fact that it treats FL clients’ local models as series of lego blocks from which the global model is built.

**Compliance With Llm Reviewing Policy:**

Affirmed.

**Final Justification:**

My concerns were mostly addressed in the rebuttal, but I still think one of my concerns remains (as explained in my rebuttal acknowledgement). Considering this, and some of the other reviewers’ remaining concerns, I am leaning towards acceptance for this paper, but am not confident/convinced enough to increase my score to a 5 (accept).

**Key Questions For Authors:**

Q1: Why didn’t you compare against any submodel extraction approaches, like HeteroFL?

Q2: Why does it make sense to use the NASWOT score here? In other words, how does maximizing this lead to better model performance?

Q3: What is the purpose of the complexity constraint (introduced in Definition 2) and how do you choose a value for it?

**Limitations:**

Yes

**Strengths And Weaknesses:**

Strengths

S1: The authors consider a nice variety of FL settings (50 clients and 100 clients, a sparse sampling scenario, IID and non-IID, etc.).

S2: Everything is well formalized and the notations are clear.

S3: The paper is generally well-written and well-organized and flows nicely.

S4: The authors do a good job, in the related work section, of explaining how their method compares to (improves upon) prior works.

S5: Tables 2, 3, 4, and 5 are very nice and clear

S6: The inclusion of multiple ablation studies is helpful in demonstrating the effectiveness of their method

S7: LEGO-FL consistently outperforms the baseline methods

Weaknesses

W1: It seems like, with stage 2, the global model is just choosing lego blocks from some clients and rejecting others. If this is true, even if the choice of blocks maximizes the performance proxy, it seems like you’d be losing information that could be important/valuable (especially in non-IID settings).

W2: The datasets used for evaluation are very simple. The paper would be a lot stronger if at least one more complicated dataset were used. For instance, some FL benchmark like LEAF that has realistic non-IID data (as opposed to restricting clients to two classes from the dataset).

W3: In stage 3, the global feature extractor is stitched with the local classifier, but it seems like the client may not be able to accommodate the global feature extractor especially in heterogeneous settings where clients are resource-constrained. This could be a detrimental limitation

W4: Only some existing methods are chosen as baselines, which is fine, but there isn’t really any justification for why the chosen baselines were chosen.

W5: It is odd that almost all of page 8, which is part of the main text, is just referencing results from the appendix. I think that many of these results would have made more sense to be included in the main text.

W6: The method seems to rely pretty heavily on NASWOT, yet the decision to use it is not justified.

W7: The text in table 1 is very tiny and should be larger.

W8: I think the computational overhead is very important to consider, but this isn’t explained very well in the paper. There are just three plots (in the appendix) that show results from one specific experiment with CIFAR-10. In the second plot, it seems like there should be three lines but I can only see 2. It would be helpful if the colors remained consistent and each line color consistently corresponded to the same method. I also don’t know what C2PR is. The paper would be much stringer if there were some theoretical proof of the algorithmic complexity instead of one empirical result.

---

> ### Author Rebuttal · Authors · 2026-03-31
>
> **Q1: [Weaknesses 1] Global Model Selection.**
> The candidate pool for global model assembly is sourced from components provided by all clients. Our evaluation favors assessing the expressivity of different blocks within diverse combinations. Thus, the optimal model is selected as the one exhibiting the highest expressivity in the per round. This implies that our method does not lose information from certain users.
>
> **Q2: [Weaknesses 2] Evaluation on Realistic Datasets.**
> We have added our experiments with two datasets from the LEAF benchmark: FEMNIST and Celeba. Experimental he setup follows the configurations in 4.1. Our results are in Table 1 (https://anonymous.4open.science/r/Supplementary-Materials-for-Rebuttal-C34D).
>
> **Q3: [Weaknesses 3&Questions 3] About Personalized Grafting.**
> We address this concern with a built--in constraint when constructing the global model. We control the generated model size by following an adaptive rule, which adds 2 layers to the maximum size of the local models. For instance, where client models range from 4 to 6 layers, we set the generation rule to a layer count in the range of 6-8. This adaptive rule accommodates heterogeneous model settings and is precisely the complexity constraint we formally define in Definition 2.
>
> **Q4: [Weaknesses 4] Baseline Selection.**
> **For the heterogeneous model task**, we selected baselines from two mainstream methods: Federated Distillation (FD) and Classifier Sharing (CS).
>   - For FD, we chose FedMD as a representative and the latest SOTA model, FedCache.
>   - For CS, we selected the representative method FedGH.
>
> Furthermore, we compared our method against baseline methods that employ similar techniques (Model Reassembly). We selected the two SOTA  methods, pFedHR and pFedClub.
>
> **For the homogeneous task**, we chose classic (FedAvg,FedProx) and recent methods (FLAYER, DisUE). The latter two were chosen because their design philosophy also involves balancing "personalization and generality". This explanation will be added to the "Baselines" of Section 4.1.
>
> **Q5: [Weaknesses 5&7] Layout and Font Adjustments.**
> We will add the content to the final version. We will increase the font size.
>
> **Q6: [Weaknesses 6&Questions 2] Decision on NASWOT.**
> Our evaluation metric is inspired by [1]. That work empirically demonstrated a high correlation between the NASWOT score and final model performance, as measured by the Kendall's Tau (τ) coefficient. Our main results also demonstrate the soundness of this choice.
>
> **Q7: [Weaknesses 8] Complexity analysis.**
> We have added a complexity analysis. This analysis will be incorporated as a standalone section.
> * **For Server-side per round**, the overhead includes three stages:
>   - **Grouping.** Extract activations for all layers (sum_i M_i <= NM) via forward passes on D_pub: O(N * |D_pub| * P_bar). Pairwise CKA on N^2 * M^2 layer pairs, each over |D_pub|/B_pub batches, costs O(N^2 * M^2 * |D_pub| * B_pub * d).
>   - **TFreeC.** Evaluate candidate architectures with one forward on a batch: O(2*N * B_pub * P_max).
>   -  **Personalized Grafting.** Fine-tuning personalized models: O(N * I_phi * B_pub * (P_G + P_C_bar)), where P_C_bar is average head size.
>  * **For Client-side, per round,** each client i performs E epochs of local training (cross-entropy + distillation). Total: O(E * sum_i |D_i| * (P_i + P_G + P_Ci)).
> * **For Communication per round,** Downlink: server sends teacher M_i (P_G + P_Ci params) to each client. Uplink: clients upload updated models (P_i). Total: O(sum_i (P_G + P_Ci + P_i)) = O(=N * (P_G + P_bar)).
>
> Regarding Figure 2(b), we acknowledge that the confusion was caused by the similar line colors, and we will use more distinct colors. We will correct the "C2PR" to "LEGO-FL".
>
> **Q7: [Questions 1] Experiments on Sub-model Extraction.**
> A key distinction is that sub-model extraction methods typically operate on models from homogeneous sources (e.g., variants of the same family), whereas our approach is designed for models from genuinely heterogeneous sources. To validate the feasibility of LEGO-FL, we have added comparisons against  HeteroFL[2], FD[3], FedRolex[4], and FedEcover[5] (Results in Table 2:https://anonymous.4open.science/r/Supplementary-Materials-for-Rebuttal-C34D).  Beyond CIFAR-10, we also benchmarked on more challenging tasks, CIFAR-100 and Tiny ImageNet. Our method consistently outperforms the baselines across these settings.
>
>  [1] Deep model reassembly[J]. NeurIPS.2022.
>
>  [2] Heterofl: Computation and communication efficient federated learning for heterogeneous clients[J]. ICLR.2021
>
>  [3] Expanding the reach of federated learning by reducing client resource requirements,ICLR.2019
>
>  [4]FedRolex: Model-Heterogeneous Federated Learning with Rolling Sub-Model Extraction.NeurIPS.2022
>
>  [5] FedEcover: Fast and Stable Converging Model-Heterogeneous Federated Learning with Efficient-Coverage Submodel Extraction.ICDE.2025

---

> > ### Author Rebuttal · Reviewer_hV7i · 2026-04-04
> >
> > Thank you for taking the time to write this rebuttal and run additional experiments. I do think my concerns were addressed well and I will maintain my positive rating of this paper.
> >
> > I am still somewhat unconvinced regarding W1. Even though all clients' components are candidates for the global model, only one combination will be selected. Even if it has the highest expressivity, I still believe there will be information loss. I don't think this is a catastrophic drawback, but the paper would be strengthened if this were better acknowledged/justified.

---

> > > ### Author Response · Authors · 2026-04-04
> > >
> > > We sincerely appreciate your continued support and constructive feedback. As recommended, we will explicitly expand the discussion on this point in the final manuscript to further strengthen our work.

---

### Official Review · Reviewer_FELJ · 2026-03-12

**Soundness:** 3
**Presentation:** 3
**Significance:** 3
**Originality:** 3
**Overall Recommendation:** 5
**Confidence:** 5

**Summary:**

Heterogeneous federated learning is a critical research topic in recent studies. This paper proposes a novel modular training paradigm aimed at "Common-to-Personalized" learning, which effectively balances shared knowledge and personalization to customize client-specific models. Specifically, the author introduces a new aggregation mechanism for heterogeneous models termed "Lego-style Reassembly". It decomposes heterogeneous networks based on functional equivalence, searches for neural blocks under sequential constraints, and reassembles them into heterogeneous models and personalized grafted models. Experimental results show the feasibility of the proposed method.

**Compliance With Llm Reviewing Policy:**

Affirmed.

**Final Justification:**

Please see the Rebuttal Acknowledgement.

**Key Questions For Authors:**

1. Comparison with NAS: "What are the distinct advantages of the proposed method compared to Neural Architecture Search (NAS)?"
2. Is there a direct correlation between the final selection of the global model and the size of the combined structure?
3. Comparison with Sub-model Extraction: "In the task of aggregating heterogeneous derivatives of the same architecture (similar to the Zoo1 setting), how does the proposed method perform compared to recent sub-model extraction techniques [1]?
[1] Liang J, Zhang L, Qu X, et al. FedEcover: Fast and Stable Converging Model-Heterogeneous Federated Learning with Efficient-Coverage Submodel Extraction[C]//2025 IEEE 41st International Conference on Data Engineering (ICDE). IEEE, 2025: 2575-2587.

**Limitations:**

yes

**Strengths And Weaknesses:**

Strengths:

1. This paper notices a critical yet underexplored problem in Heterogeneous Federated Learning (HFL): how to customize personalized models that balance common knowledge and personalization for clients with diverse architectures, which has been overlooked in existing HFL works.
2. The proposed method advances the modeling paradigm from neuron-level pruning to functional block-level modeling. This shift accommodates diverse heterogeneous architectures and mitigates the limitations of existing modular approaches regarding local personalization.
3. Experiments are conducted across settings involving significant statistical heterogeneity, heterogeneous model distributions, and large-scale client participation. The results provide empirical support for the effectiveness of the proposed approach.
4. The authors provide open-source code for the proposed training paradigm, which facilitates reproducibility and benefits future research.

Weakness:

1. Insufficient analysis of the core methodology. The overall concept shares similarities with Neural Architecture Search (NAS). However, the manuscript lacks a discussion on the feasibility of directly applying existing NAS methods to this setting. Furthermore, the unique advantages of the proposed framework over standard NAS solutions are not clearly articulated.
2. Insufficient experimental verification. While the paper claims distinct advantages over sub-model extraction techniques, there is a lack of experimental results to support this.

---

> ### Author Rebuttal · Authors · 2026-03-31
>
> **Q1: [Weaknesses 1] Fundamental Difference from Standard NAS.**
> Standard NAS methods aim to find a single, optimal-performance model，and are inherently unable to provide personalized models. In contrast, our method assembles a global model from pre-trained components and then performs personalized grafting. This allows us to provide models that balance both generality and personalization.
>
> **Q2: [Weaknesses 2] Adding Comparison with Modular Methods.**
> Thank you for the practical suggestion! Following the same experimental setup as in the main text, we have added a direct performance comparison between LEGO-FL and FedEcover under the Model Zoo 1 setting. The results of this supplementary experiment are presented in the table below.
>
> | Model Zoo | Method | MNIST (IID) | MNIST (NON-IID) | CIFAR-10 (IID) | CIFAR-10 (NON-IID) | SVHN (IID) | SVHN (NON-IID) |
> |:---|:---|:---:|:---:|:---:|:---:|:---:|:---:|
> | 4 lightweight CNNs | FedEcover | 94.59 ±0.70 | 92.34 ±1.54 | 71.41 ±0.22 | 61.60 ±1.56 | 87.98 ±0.72 | 84.06 ±1.64 |
> | 4 lightweight CNNs | **LEGO-FL (Ours)** | 97.25 ±0.73 | 95.47 ±1.55 | 80.82 ±1.82 | 78.96 ±1.31 | 87.70 ±1.93 | 86.69 ±1.20 |
>
>
> **Q3: [Weaknesses 3] Global Model Selection.**
> Based on empirical observation, evaluating the global model solely on its layer count is limiting. The final choice of the global model architecture is determined by its intrinsic mathematical properties—specifically, its expressivity.

---

> > ### Author Rebuttal · Reviewer_FELJ · 2026-04-02
> >
> > I thank the authors for their effort and careful rebuttal. The responses sufficiently resolve my questions, and I will keep my initial positive evaluation.

---

### Official Review · Reviewer_tYry · 2026-03-12

**Soundness:** 3
**Presentation:** 4
**Significance:** 3
**Originality:** 3
**Overall Recommendation:** 4
**Confidence:** 3

**Summary:**

This paper proposed a new framework for heterogenous federated learning called LEGO-FL. The main idea is to allow
allow more diverse client architectures by enabling modular construction of models. The framework proceeds in three stages:
 - Architecture generation:  Clients send their models to the server, which decomposes them into equivalence sets of "lego blocks" that are semantically similar and performs similar tasks. Similarity is measured using Centered Kernel Alignment which finds semantically similar blocks by checking their activations. This returns a similarity matrix that tells how the functional similarity of specific parts of the model.  The author then uses Affinity propagation for clustering blocks that perform similar tasks. Once semantically similar blocks are identified, the next step is to build a Candidate global model. This is done by a novel algorithm proposed by the authors called CT search. Instead of going through the entire search space by starting at a random starting piece, CTsearch instead uses the cluster centers found earlier as their starting block. They then build upon this by only adding blocks in a predetermined semantic order.

- Choosing optimal global model: The next step is to get the optimal global model from the candidates generated earlier. The paper does this without training by using model expressivity (NASWOT scores) as the criterion.

- Updating local client models: After selecting the optimal global model, the next step is to update the local client models. The paper does this by distilling knowledge from the global model to the children. They build a teacher network by taking the head of the global model (which captures semantic information) and stitches it with the tail of the local model (which captures client specific information). They then use distillation to transfer knowledge from the teacher to the local model.

**Compliance With Llm Reviewing Policy:**

Affirmed.

**Final Justification:**

The paper proposes a novel method for model heterogeneous federated learning based on decomposing models into functionally equivalent sets. The paper is well written, and the authors evaluate their approach across several different federated settings, demonstrating strong empirical performance.

In their rebuttal, the authors have clarified their choice of using NASWOT as the evaluation criterion and have provided additional complexity analysis. I am satisfied with their rebuttal and will be maintaining my current score.

**Key Questions For Authors:**

1) How is the semantic ordering $\pi(l)$ determined? How do you decide which block to add first? Once you form clusters of similar performing blocks, how do you decide which cluster does the feature extraction and other early stage tasks and which ones do the later stage tasks such as extracting finer details?

2) Expressivity is used as a criterion to select the optimal global model without training. However, it is not clear to me why expressivity is a good criterion for this. I understand that you do not typically have a validation set on the server side since that violates the privacy issues involved in federated settings. Can you provide some intuition as to why you chose expressivity as the criterion? Is it based on empirical results as mentioned in the paper when testing using other metrics or is there some theoretical justification for it?

**Limitations:**

yes

**Strengths And Weaknesses:**

- The paper proposes a new framework for model heterogeneity in federated learning. The idea of decomposing models into lego blocks and clustering them based on their functional similarity is interesting and seems to be a good way to identify semantically similar blocks.

- The CT search algorithm makes the process of building candidate models more efficient by avoiding random searches. However, the exact semantic ordering they use is a bit unclear.

- The paper has conducted several experiments to evaluate the proposed method and the results show that it performs well compared to other baselines. They test their model on different levels of heterogeneity with 50 clients. They also try to test the scalability of the model by testing it on 100 clients (albeit it still seems small compared to realistic settings). They also perform an ablation study to show the importance of each component and the performance increase.

- There are a few sections of the paper that refers to prior works but do not mention the specific papers. For example, in section 3.3, the paper mentions "Prior works rely on training based evaluation or fine-tuning adaptation layers" but does not mention the specific papers.

- The paper does not talk about the efficiency of the proposed algorithm in detail. They provide experiments that compare with other baseline model reassembly methods but a brief discussion about the efficiency of the method and how it scales with the number of clients and the size of the models would be helpful.

---

> ### Author Rebuttal · Authors · 2026-03-31
>
> **Q1: [Weaknesses 1& Key Questions 1] Details of the CT Search ordering**.
> Thanks for your critical point. Our search algorithm follows a global semantic order. We build a global set of blocks by traversing all non-repeating neural blocks in the model zoos. Then, we sort this global set according to the built-in rules of neural architectures, generating a global ordering map. Therefore, the first block is typically the convolutional layer with the smallest block index, and the early and late-stage tasks are sorted by placing feature extraction blocks before classification blocks.
>
> **Q2: [Weaknesses 2] Adding Citations**.
> In Section 3.3, the idea of repeatedly training the generated model and then matching it to clients, or using it as the final optimal model, is discussed. The former approach can be seen in[1,2]. The latter approach is seen in [3,4].
>
> **Q3: [Weaknesses 3] Efficiency of the Method**.
> We have added a time complexity analysis.
> * **For Server-side, per round**, the overhead mainly includes three stages:
>   - **Adaptive Semantic Grouping.** Extract activations for all layers (sum_i M_i <= NM) via forward passes on D_pub: O(N * |D_pub| * P_bar). Pairwise CKA on N^2 * M^2 layer pairs, each over |D_pub|/B_pub batches, costs O(N^2 * M^2 * |D_pub| * B_pub * d).
>   - **TFreeC.** Evaluate candidate architectures with one forward on a single mini-batch: O(2*N * B_pub * P_max).
>   -  **Personalized Grafting.** Fine-tuning personalized models: O(N * I_phi * B_pub * (P_G + P_C_bar)), where P_C_bar is average head size.
>  * **For Client-side, per round,** each client i performs E epochs of local training (cross-entropy + distillation). Total: O(E * sum_i |D_i| * (P_i + P_G + P_Ci)).
> * **For Communication per round,** Downlink: server sends teacher M_i (P_G + P_Ci params) to each client. Uplink: clients upload updated models (P_i). Total: O(sum_i (P_G + P_Ci + P_i)) = O(=N * (P_G + P_bar)).
> * **Scalability with Client Number $N$.** The server-side complexity contains a term from the pairwise CKA computation. While quadratic in theory, the clustering process is fast in practice, as shown in Figure 2(a).
> * **Scalability with Model Size.** The per-round complexity contains terms that scale linearly with model parameters in several components: server-side feature extraction, client-side training, TFreeC evaluation, and Personalized Grafting. These terms all grow linearly with the size of the models involved, which is consistent with standard FedAvg.
>
> **Q4: [Weaknesses 4] Basis for Choosing Expressivity.**
> Our 'expressivity' criterion is inspired by [5]. That work systematically tested 8 training-free metrics across numerous image classification tasks and found that the NASWOT score exhibits a high correlation with final model performance (represented as transferability in their study), as measured by the Kendall's Tau (τ) coefficient.
>
> **References**:
>
>  [1] Towards personalized federated learning via heterogeneous model reassembly[J]. NeurIPS.2023.
>
>  [2] pFedClub: Controllable heterogeneous model aggregation for personalized federated learning[J]. NeurIPS.2024.
>
>  [3] Big transfer (bit): General visual representation learning. In European conference on computer vision, Springer, 2020
>
>  [4] How to train your ViT? Data, Augmentation, and Regularization in Vision Transformers[J]. Transactions on Machine Learning Research.2022.
>
>  [5] Deep model reassembly[J]. NeurIPS.2022.

---

> > ### Author Rebuttal · Reviewer_tYry · 2026-04-03
> >
> > I thank the authors for their detailed rebuttal and appreciate the added complexity analysis. The rebuttal sufficiently clarifies my questions and I will maintain my current score.

---

### Decision · Program_Chairs · 2026-04-30

**Decision:**

Accept (regular)

**Comment:**

This paper notices a critical yet underexplored problem of how to customize personalized models that balance common knowledge and personalization for clients with diverse architectures, which has been overlooked in existing HFL works. The proposed method advances the modeling paradigm from neuron-level pruning to functional block-level modeling. The results provide empirical support for the effectiveness of the proposed approach.